# The Evaluation of a New ELISA-Based Kit for Total Microcystins as an Early Detection Tool for Microcystin Blooms in Source Waters and Its Application State-Wide to Oregon Source and Finished Drinking Waters

**DOI:** 10.3390/toxins17020053

**Published:** 2025-01-24

**Authors:** Katie Adams, Kale Clauson, William A. Adams, Rochelle G. Labiosa, Theresa McBride, Aaron Borisenko, Stuart W. Dyer, Ned Fairchild, Barry V. Pepich

**Affiliations:** 1Region 10 Laboratory, U. S. Environmental Protection Agency, 7411 Beach Dr E., Port Orchard, WA 98366, USA; 2Oregon Department of Environmental Quality, 700 NE Multnomah St., Suite 600, Portland, OR 97232, USA; 3Office of Ground Water and Drinking Water, U.S. Environmental Protection Agency, 26 W. Martin Luther King Dr., Cincinnati, OH 45268, USA; 4Region 10 Water Division, U.S. Environmental Protection Agency, 1200 6th Ave, Seattle, WA 98101, USA; 5Oregon Health Authority, 500 Summer St. NE, Salem, OR 97301, USA

**Keywords:** ELISA, drinking water, reference method, microcystin, cyanobacteria

## Abstract

Due to cyanobacterial toxin (cyanotoxin) contamination issues in 2018, the city of Salem, Oregon, issued a 33-day do-not-drink advisory for vulnerable people among the 200,000 residents. After the incident, the state of Oregon put in place drinking water rules to require the routine testing of raw water, as well as finished water, in cases where the raw water cyanotoxin concentrations exceeded trigger values. The United States Environmental Protection Agency (EPA) total microcystins drinking water health advisory level (HAL) for small children is 0.3 µg/L. This is equivalent to the minimum reporting level (MRL) for EPA Method 546. Consequently, there was no ability to provide early warnings via toxin testing for total microcystins using the EPA method. In this study, we performed a comparison of the precision and accuracy of the enzyme-linked immunosorbent assay (ELISA) described in the EPA method to a more sensitive assay, the Streptavidin-enhanced Sensitivity (SAES) assay. Based on these precision and accuracy studies and quantitation limit determinations and confirmations, the EPA Office of Ground Water and Drinking Water (OGWDW) has concluded the SAES kit meets the requirements of EPA Method 546. With an MRL that is one-third of the original concentration, the new kit provides a small but critical window for identifying early warnings. Challenges remain with providing early warnings due to the variability in bloom dynamics; however, the new MRL allowed Oregon to lower the trigger level for susceptible systems, thereby providing an additional early warning.

## 1. Introduction

Carmichael [1] traced the first scientific report on the acute poisoning of animals from consuming water containing toxin-producing cyanobacteria or cyanotoxins to a paper by Francis [2], published in *Nature* over 130 years ago. Since then, a much better understanding of these organisms and the toxins they produce has emerged. Today, many US states have implemented monitoring and advisory programs to detect and track harmful cyanobacterial blooms (HCBs), the umbrella term for all harmful blooms, including algae/algal biotoxins and cyanobacteria/cyanotoxins, in recreational and drinking waters [3]. Conditions leading to cyanotoxin production can be influenced by many environmental factors, including warmer water temperature, nutrient levels (nitrogen and phosphorus), and slower flow/increased stratification. Eutrophication due to increased urbanization, agriculture, forestry, wastewater, onsite sewage discharges, and other sources is often associated with cyanotoxin increases [4]. Increased average summer temperatures and droughts that lead to significant wildfire events, and other changes in runoff patterns that have been identified in the Pacific Northwest, can also lead to additional nutrient loading into watersheds [5,6]. These factors, together with reduced riparian vegetation, could further increase the likelihood of HCBs. Harmful cyanobacterial algal blooms, which can be a year-round phenomenon in southern US states and seasonally cyclical in northern US states, are common nationwide [7].

Local and US state agency-issued freshwater HCB advisories (2015–2021) have been collated and summarized by the United States Environmental Protection Agency (EPA), and the total number of advisories by month and year displays an increasing trend from 2015 through 2021 [8]. This trend could be due to an increase in awareness, an increase in cyanotoxin events, or a combination of the two, but the total number of advisories issued annually in this period indicates that cyanobacterial HCB issues are a ubiquitous and resource-intensive water quality management issue. A total of over 6000 monthly tallied advisories were identified across the country in 2015–2021, with approximately 800 identified in the northwest US states of Idaho, Oregon, and Washington. One class of cyanotoxins, the microcystins (MCs), are hepatotoxins of varying toxicity levels comprising more than 200 congeners reported to date [9]. Microcystins are widespread in occurrence and have been associated with numerous global incidents [10,11,12]. The ability of this family of cyanotoxins to inhibit protein phosphatases gives rise to additional concerns that sublethal doses may contribute to cancer [13]. In addition to state- or local-issued advisories listed above, data for MCs have been collected for a national random sample of lakes approximately every five years since 2007 as part of the EPA’s National Lakes Assessment (NLA). According to NLA data (from 2007, 2012, and 2017), approximately one out of three lakes sampled in any given year had detectable concentrations of “total” microcystins (MCTs), which accounts for all MC congeners, on the day of sampling [7,14]. In 2015, the EPA established a ten-day health advisory level (HAL) for microcystin-LR in drinking water of 0.3 µg/L and 1.6 µg/L for young children (infant to pre-school) and adults, respectively [15]. In a recent study on the toxicity of 10 MC congeners administered orally in mice, researchers noted a wide range of toxicities, with the highest toxicity levels found for MC-LA and MC-LR. In addition, MC-LA was found to induce serum alterations resulting in jaundice [16], which was not noted for MC-LR at the administered dose. Today, many US states are using the EPA HAL as a reference point for making public health decisions in source and finished waters.

MCs were included in the first Drinking Water Contaminant Candidate List (CCL), a list of drinking water contaminants that are known or anticipated to occur in public water systems and are not currently subject to EPA drinking water regulations, as “Cyanobacteria (blue-green algae), other freshwater algae, and their toxins”; selected MCs, as well as an MCT result, were surveyed in the fourth cycle of the Unregulated Contaminant Monitoring Rule—UCMR 4 [17,18]. The EPA published the first UCMR in 1999 [19] following the 1996 amendments to the Safe Drinking Water Act (SDWA). Under the UCMR, the EPA conducts national surveys of finished drinking waters to investigate the occurrence of up to 30 contaminants from the CCL in each cycle. Upon completion of the survey, the agency uses the occurrence data for decision making in their regulatory determination process. UCMR 4, in which monitoring was concluded in 2021, surveyed approximately 6000 public water systems, including all large water systems serving over 10,000 people [18].

In UCMR 4, two analytical methods were employed for MC quantification—EPA Method 546 and EPA Method 544. The first, EPA Method 546, is an enzyme-linked immunosorbent assay (ELISA) technique based on a method first reported by Fischer et al. [20]. This assay is based on an antibody directed against the amino acid ADDA (4E,6E-3-amino-9-methoxy-2,6,8-trimethyl-10-phenyldeca-4,6-dienoic acid), which is common to most of the penta- and heptapeptide toxin congeners. The resulting EPA method had both high cross-congener reactivity and sufficient sensitivity for an MCT and nodularin measurement, which made it a suitable choice for UCMR 4 [21]. Water samples that yielded an analytical result for MCTs above the minimum reporting level (MRL) were required to be analyzed with the more specific and more sensitive liquid chromatography–tandem mass spectrometer method, EPA Method 544 [22]. Due to the availability of analytical standards at the time of development, EPA Method 544 was limited to six MC congeners and nodularin-R and was intended to provide congener-specific information for the survey. EPA Method 544 was not necessarily expected to be confirmatory to EPA Method 546 due to measurement differences between the methods. The MRL used in UCMR 4 for EPA Method 546 was 0.3 µg/L, which was based on a statistical analysis of the analytical capability of the method. This MRL—equivalent to the EPA HAL for young children—was deemed adequate for the purposes of the survey.

Many municipalities monitor source and influent surface waters for the onset of algal toxin blooms and have already implemented protective measures to respond to toxins during treatment when detected. The appropriate selection of cyanotoxin treatment technologies requires knowledge of growth patterns and species of the cyanobacteria, whether the toxins are intracellular or extracellular, and the duration and magnitude of HCB events [9,23,24]. Having an EPA screening method for MCTs with enhanced sensitivity is highly desirable as an early warning tool for municipalities managing cyanotoxin issues.

In this paper, we report a validation study of EPA Method 546 using a new commercially available ELISA kit for measuring MCTs and nodularin, which also uses an antibody directed against the ADDA moiety. This validation study was structured to determine whether the new kit conforms to the precision and accuracy requirements of EPA Method 546. We also assessed the potential for enhanced sensitivity with the new kit as an early-warning tool for the onset of toxigenic HCBs. Finally, we report the performance of this method using the new kit, including the occurrence data, analyzed by Oregon Department of Environmental Quality (ODEQ) during the 2022 monitoring cycle of susceptible drinking water sources state-wide, in keeping with Oregon’s cyanotoxin monitoring and public notification at public drinking water systems, Oregon Administrative Rule (OAR) (OAR 333-061-0510 through 0580), which was implemented in 2018 to protect public health [25].

## 2. Results and Discussion

EPA Method 546 utilizes a commercially available ELISA test kit, and in the present study, both participant labs had automated ELISA instruments available. These automated systems are more suitable for laboratories seeking higher throughput and/or rapid turnaround, and they also automate the timing of reagent addition, which is critical to method precision and accuracy. In contrast, EPA Method 546 was initially validated by EPA using a non-automated, manually performed bench assay.

The EPA Method 546 immunochemistry test kit is based on the original work of Fischer et al. [20]. The manufacturer’s product insert lists a calibration range from 0.15 to 5.0 µg/L and a Minimum Detection Limit (MDL) of 0.10 µg/L [26]. The test kit evaluated in this study differs from the EPA Method 546 kit in that it uses a streptavidin–biotin interaction to conjugate the detection and reporting antibodies, and it is reported to have a calibration range from 0.05 to 5.0 µg/L and an MDL of 0.016 µg/L [27]. It is worth noting that the manufacturer’s stated MDL is the calculated MCT concentration at 90% B/B_0_ based on the average of 46 standard curves (B/B_0_ is obtained with the given sample absorbance divided by the maximum absorbance obtained with a blank). The chemistries of the SAES kit were initially determined by the EPA to be sufficiently different from the kit listed in EPA Method 546, and the SAES kit was considered to be outside of acceptable method-prescribed flexibility. However, EPA Method 546 does allow for the use of equivalent ADDA ELISA test kits. Validation studies were conducted to determine method test kit equivalency between the streptavidin-enhanced kit and the kit listed in EPA Method 546.

### 2.1. Method Equivalency Study

Prior to analyzing any samples, both labs confirmed their calibrations for the SAES test kits according to the EPA quality control requirements in EAP Method 546 using independent second source standards [21]. Each lab had acceptable average recoveries (70–130% recovery). Because the analyte concentration is inversely proportional to absorbance in this ELISA technique, blanks return a high absorbance. Poor blank values can be caused by several factors, including poor precision in the calibration curve. Blanks were processed through the entire method, including the addition of the sodium thiosulfate dechlorinating agent. Blank results are reported with the quality control standards and were acceptable for all analyses (<½ the MRL). These data are summarized in Table 1, original data see Appendix A.

Precision and accuracy for reagent water and finished tap water were determined for both labs at four concentrations using seven replicates and are presented in Table 2 and Table 3, respectively. Precision and accuracy at the two higher concentrations were comparable between the standard test kits described in EPA Method 546 and the SAES test kits. Both labs achieved recoveries for the SAES kits in the range of 70–130% with RSDs ≤ 15%.

Because regulatory decisions that have cost and human health impacts are made in part using UCMR occurrence data, the EPA developed a new quantitation level concept, the lowest concentration minimum report level (LCMRL), that incorporates both precision and accuracy in analytical measurements [28,29,30]. LCMRL data are collected from multiple laboratories during UCMR development and are used to determine the MRLs for that cycle. An additional calculation, the EPA-defined MDL, which is used extensively in other EPA method compendiums, is based on low-level method variance and blank performance [31]. Both labs determined the MDL and the LCMRL using the appropriate EPA procedures [28,30]. The detection limits for the SAES kits were 0.027 µg/L and 0.010 µg/L, and LCMRLs were 0.21 µg/L and 0.09 µg/L for Labs 1 and 2, respectively.

Because LCMRL determinations require a significant number of replicates processed through the entire method, the EPA developed an alternative procedure for labs to demonstrate sufficient precision and accuracy at a specific MRL concentration with seven replicates [28,32]. Both labs processed the data from Table 2 through the MRL confirmation test (see Section 9.1.3.3 in EPA Method 546 [21]) and obtained similar results. Both labs were able to confirm MRLs above the LCMRL at 0.115 µg/L, but failed to achieve confirmation below the LCMRL at 0.075 µg/L. These data are presented in Table 4.

From these data, it is evident that the new SAES kit performs as well as the original EPA Method 546 test kit, and offers additional sensitivity with robust precision and accuracy down to 0.115 µg/L. We note that during both evaluations, the LCMRLs determined in this study were above the detection limit reported by the manufacturer. Based on these data, the EPA OGWDW concluded that the new test kit meets the requirements of EPA Method 546 and may be used as an equivalent ADDA ELISA test kit in the method.

### 2.2. Method Performance in Oregon Municipal and Source Waters

In the summer of 2018, MCTs were detected at levels of concern for vulnerable populations in the drinking water system of Oregon’s state capital, Salem. A drinking water advisory for vulnerable people was issued by the City of Salem, which remained in effect for 33 days. This led to the development of emergency rules to protect public water systems in Oregon from two cyanotoxins: MCT and cylindrospermopsin (CYN), which required public water facilities identified as potentially susceptible to cyanobacteria blooms to routinely test for MCTs and CYN and notify the public about the test results. In 2018, over 100 facilities were required to monitor under the emergency rule between July and October. In December of 2018, permanent rules were adopted, requiring susceptible facilities to monitor for MCTs and CYN at raw water sampling points from May 1 to October 31. Subsequent data and risk analysis were used to select 54 susceptible public water facilities for testing under the permanent rule in 2019. The list has since increased to 65 facilities, in 2023.

Table 5 shows the method performance for the QC samples analyzed during the 2022 season. These analyses included samples from surface waters (recreational HCB response samples), source waters and finished waters fortified at the mid-point of the calibration range. The QC samples also included laboratory-fortified blanks and quality control samples. Precision and accuracy in reagent water and the fortified matrices were comparable based on one-way analysis of variance (n = 68, mean recovery = 92.0 ± 15.9, *p* < 8 × 10^−7^, f-value = 10.62), and the method was found to have performed acceptably in all matrices analyzed (i.e., finished, source, surface, and ground waters).

We present the results from the Oregon monitoring program by month over a three-year period spanning 2020–2022 in Figure 1. For the 54 public water sources monitored during this period, no detections were found in finished drinking water. Utilizing the new SAES chemistry, we found that the largest proportion of quantifiable MCT detections fell below the HAL during the earlier summer months (Figure 1). The number of water systems with MCT concentrations above the HAL peaked in the month of September during this three-year monitoring period. Due to the increased sensitivity of the SAES kits, under the new OAR (requiring additional monitoring when a trigger level of 0.20 µg/L is exceeded in raw water), ten weeks of additional sampling would have occurred in 2020–2021 across four public water systems. This additional sampling, which accounts for sampling already occurring due to exceedance of the 0.3 µg/L HAL, would have given these facilities up to a week of early warning, allowing the facilities to adapt treatment if necessary to manage these events. EPA has published a suite of resources geared towards treatment optimization for removing cyanotoxins, and what changes might be necessary to adopt based on toxin characteristics and concentrations (EPA 810-B-16-007).

The number of source waters that exceeded the DL during the monitoring years is shown in Figure 2. For each source, only the maximum MCT concentration detected during the annual cycle is reported. Sorting the groups by year can show whether a severe season occurred statewide or whether it was limited to a few water systems. While the number of water systems sampled was the same in each year, the number of systems where MCTs were detected at any level increased from eight in 2020 and seven in 2021 to twelve in 2022, primarily due to the lowered quantification limit provided by the SAES kits. However, 2021 had the most public water systems with MCT detections over the HAL (n = 5) and 2022 had the fewest (n = 3). For 2022, 75% of the water sources with quantifiable MCT concentrations would not have been identified or reported with EPA Method 546 or the HAL of 0.3 µg/L established under the previous OAR.

## 3. Conclusions

The Oregon Administrative Rule (OAR 333-061-0510 through 0580), enacted by the legislature in the aftermath of the Salem drinking water incident, called for bimonthly testing of surface water sources of drinking water that had any recorded cyanotoxin activity or history of HCBs. The raw water was tested at the point of entry to the treatment facility. If any cyanotoxins were detected above the HAL in raw water, an immediate sample in the finished water was collected. This sampling frequency was deemed adequate; however, the sensitivity of the ELISA method (EPA Method 546) was of concern. The trigger limit for additional testing initially set by the OAR, 0.3 µg/L for MCTs, was the same as the HAL for vulnerable populations. Without a method for MCTs with additional sensitivity, Oregon was concerned about potential scenarios where a bloom would skirt the routine testing window and only be detected when toxin concentrations were already at, or exceeding, health-based thresholds.

This study has shown that EPA Method 546 using the new ELISA SAES kits is able to detect MCTs at one third of the MRL achievable by EPA Method 546 using the kit listed in the method, and that it has comparable precision and accuracy. Based on this performance, the EPA determined this kit to meet the requirements of EPA Method 546 and be applicable as an equivalent test kit in the method. Method performance during 2022 across a wide range of recreational, source, and finished waters in Oregon was also acceptable. When this new test kit was applied to Oregon’s cyanotoxin monitoring program over a three-year period, data illustrated the benefits that a more sensitive method offers in terms of early warnings for public water systems (Figure 1).

Although it is difficult to predict the trajectory in toxicity of a given HCB event, the increased sensitivity of the SAES kit provides at least some opportunity for facilities to adjust treatment parameters prior to exceedance of the HAL in raw water for MCTs. Following the demonstration of equivalency of the SAES kit, the OAR has been amended and the trigger level for additional testing has been lowered to 0.20 µg/L for MCTs. This update takes advantage of the increased sensitivity and therefore improves the likelihood that a bloom will be detected in the source water before the onset of harmful MCT levels. The increase in sensitivity also provides reassurance that toxins are removed during the treatment process of finished drinking water. Early detection of bloom onset and toxin production also provides a better picture of the cycle of these blooms in Oregon waterbodies, while also reducing the risk to human and animal health through earlier detection.

As the circumstances leading up to the drinking water advisory in Salem, and the aftermath of that event demonstrate, state programs and local water providers benefit from the early detection of HCB events and changes in toxin production when toxin concentrations are approaching HALs. This method using the SAES ELISA kit offers an additional tool for consideration nationally that achieves a modest improvement in sensitivity as it relates to MCT quantification and has been evaluated for equivalency by the EPA and determined acceptable. With appropriate early warnings, treatment modifications can be enacted to preclude breakthroughs into finished water and prevent health risks to the public. Water providers and health authorities alike must have access to sensitive and accurate information to protect human health.

## 4. Materials and Methods

### 4.1. Environmental Sample Collection

Sampling kits were prepared by the Oregon DEQ laboratory and shipped to participating public water systems. Each sampling kit contained a sample bottle, four 6 oz. ice packs, a bubble wrap bottle sleeve, sample bottle labels, a chain of custody form, and prepaid return shipping labels inside of a Styrofoam shipping cooler. Prior to shipment, a tablet containing 10 mg of sodium thiosulfate (Brim Technologies #T10, Eatontown, NJ, USA) was added to each 125 mL amber glass collection bottle with a PTFE (Polytetrafluoroethylene)-lined screw cap and a label affixed with a mark corresponding to the 100 mL fill line. Oregon public water systems flushed the sampling taps for 5 min, filled each bottle to the fill line, and capped and thoroughly mixed the sample and chlorine quenching agent. Upon receipt at the laboratory, sample temperatures were verified to confirm that they were at or below 10 °C. Finished water samples were checked for the presence of residual chlorine upon receipt using a colorimetric DPD (N,N-diethyl-p-phenylenedi-amine) technique in keeping with the EPA method requirements.

### 4.2. Sample Preparation and Analysis

The Oregon DEQ laboratory prepared field samples and batch quality control samples by transferring 5 mL aliquots of samples and standards into clear 20 mL borosilicate glass vials with PTFE-lined screw caps for the freeze–thaw process (Thermo Scientific (Waltham, MA, USA) #B7800-20; Type 1, Class A). Three freeze–thaw cycles were performed by freezing racks of vials at an angle in a −20 °C freezer until no liquid water was visible in the samples (ca. 60–90 min), then fully thawing in a reciprocating water bath at 30 °C (ca. 5–10 min). After thawing, samples were agitated by vortexing for ca. 10 s. After the cycles were completed, samples were decanted into a syringe body with the plunger removed (HSW™ (Tuttlingen, Germany) Norm-Ject™ #ABC5LS) and a new syringe filter affixed to the Luer fitting (PVDF (Polyvinylidene)) membrane with a glass fiber prefilter, 0.45 μm pore size, Millipore (Burlington, MA, USA) Millex™ HPF #SLHVM25). The plunger was reinserted, and the first 3 mL of filtrate was discarded before collecting the remaining filtrate in a clean 4 mL amber borosilicate glass vial with a PTFE-lined screw cap (Thermo Scientific #C4015-17AW).

The EPA laboratory preparation was similar, with the following exceptions. A 10 mL aliquot of sample was transferred into a 20 mL amber borosilicate glass vial with PTFE-lined screw caps (Environmental Sampling Supply (San Leandro, CA, USA) part# 0020-0400-PC). During the freeze–thaw cycles, the samples were thawed at ambient room temperature, and the samples were not agitated by vortexing between cycles but only after the 3 freeze–thaw cycles were completed. For filtering, samples were aspirated into a disposable 5 mL syringe with a Luer adapter (Thermo Scientific part# S7515-5), and a new syringe filter (Environmental Express (Charleston, SC, USA) part SF012G, 25 mm glass fiber filter with a 1.2 µm pore size) was attached. The first 5 mL was filtered to waste, and the remaining the 4 mL was transferred to amber borosilicate glass vials with PTFE-lined screw caps (Kimble (Vineland, NJ, USA) part# 60912B-1).

Both laboratories used a Cyanotoxin Automated Assay System (CAAS), currently manufactured as Awareness Technologies (Palm City, FL, USA) Model 2910, to process ELISA plates using Abraxis Manager 6.4.1.1171 software. The test kits being compared in this study were Gold Standard Diagnostics (Davis, CA, USA) Microcystins/Nodularins (ADDA) (EPA Environmental Technology Verification) (EPA Method 546) ELISA (P/N 520011) and Gold Standard Diagnostics Microcystins/Nodularins (ADDA) SAES ELISA (P/N 520011SAES).

EPA Method 546 requires the use of a congener-independent indirect competitive ELISA to detect the ADDA moiety of microcystins and nodularin. In this assay, MCTs in the sample and antigen immobilized on the wells compete for the binding sites of a primary detection antibody in the solution. After washing, an enzyme-conjugated (horse-radish peroxidase (HRP)) secondary antibody is then added to the wells, which binds the primary antibody during an incubation step. Following a second wash step, a colorizing substrate, tetramethylbenzidine, is added to the wells to develop a signal via the HRP-mediated reaction that is inversely proportional to the concentration of ADDA in the sample. After a set incubation time, color generation is stopped by adding a dilute acid solution. Finally, the absorbance (λ = 450 nm) of each well is measured. The concentration of MCTs is calculated using a four-parameter logistic calibration curve. The SAES MCT assay uses the same ADDA-specific primary antibody which has been biotinylated and a streptavidin–HRP conjugate.

The fortified materials used to evaluate performance at different concentrations were created from Gold Standard Diagnostics Microcystins/Nodularins (ADDA) spiking solution MC-LR (50 µg/L (>95% purity; HPLC, 238 nm)). All materials were diluted in larger volumes and subsampled to create multiple identical replicates. Dilutions were made in either reagent water (>18 MΩ resistivity) or tap water dechlorinated with sodium thiosulfate (CAS 7772-98-7). All test materials were processed as described in EPA Method 546, except that when filtering as described in EPA Method 546—Section 11.2 [21], the first 3–5 mL of the sample was filtered to waste as noted above.

## Figures and Tables

**Figure 1 toxins-17-00053-f001:**
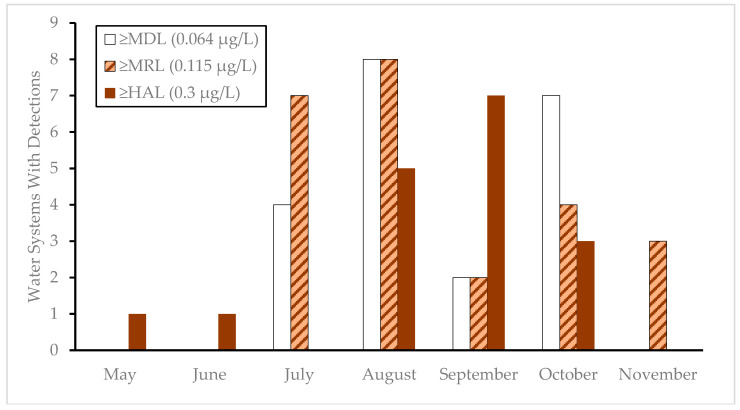
Monthly distribution of Oregon public water system source waters with microcystins detected at or above threshold limits for 2020–2022.

**Figure 2 toxins-17-00053-f002:**
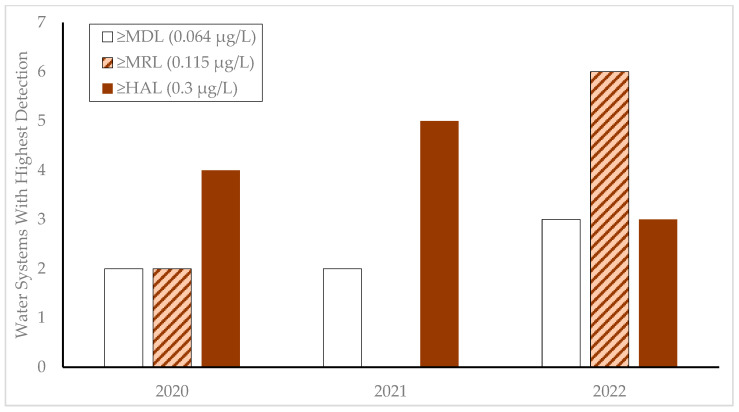
Annual count of Oregon public water systems for binned concentrations (>MDL, >MRL, >HAL) of microcystins detected in municipal source waters during the 2020–2022 monitoring years.

**Table 1 toxins-17-00053-t001:** Initial quality control (QC) confirmation for EPA Method 546 using SAES-ELISA kit.

QC Measure	Lab 1	Lab 2
Blank Determination ^1^	0.0036 µg/L	0.0087 µg/L
Quality Control Standard ^2^	104 ± 0.73 (%Rec)	85.4 ± 2.1 (%Rec)

^1^ Blank values are based on an average of five replicates. ^2^ QCSs, independent second source standards, were prepared and analyzed in triplicate at 0.75 and 0.30 µg/L for Lab 1 and Lab 2, respectively.

**Table 2 toxins-17-00053-t002:** Comparison of precision and accuracy in reagent water for EPA Method 546 using standard ELISA kit and SAES-ELISA kit.

Fortified Concentration	Standard ELISA Kit (%Rec)	SAES-ELISA Kit Lab 1 (%Rec)	SAES-ELISA Kit Lab 2 (%Rec)
0.50 µg/L	115 ± 5.4	106 ± 10	98.2 ± 7.2
0.30 µg/L	105 ± 8.8	113 ± 7.1	not evaluated
0.115 µg/L	-	98.3 ± 9.0	93.8 ± 5.9
0.075 µg/L	-	106 ± 15	89.5 ± 14

**Table 3 toxins-17-00053-t003:** Comparison of precision and accuracy in finished tap water for EPA Method 546 using standard ELISA kit and SAES-ELISA kit.

Fortified Concentration	Standard ELISA Kit (%Rec)	SAES-ELISA Kit Lab 1 ^1^ (%Rec)	SAES-ELISA Kit Lab 2 ^2^ (%Rec)
0.50 µg/L	107 ± 5.1	100 ± 10	109 ± 5.0
0.30 µg/L	117 ± 12	105 ± 12	117 ± 6.4
0.115 µg/L	-	102 ± 7.4	119 ± 8.5
0.075 µg/L		103 ± 6.3	not evaluated

^1^ Drinking water for Lab 1 was local chlorinated drinking water from a ground water source. ^2^ Drinking water for Lab 2 was local chlorinated drinking water from a surface water source.

**Table 4 toxins-17-00053-t004:** MRL confirmation results.

Fortified Concentration	Lab	Upper_PIR_ ^1^	Limit	Lower_PIR_ ^1^	Limit
0.50 µg/L	1	147%	Pass	65%	Pass
0.50 µg/L	2	126%	Pass	70%	Pass
0.30 µg/L	1	141%	Pass	85%	Pass
0.30 µg/L	2	146%	Pass	87%	Pass
0.115 µg/L	1	134%	Pass	63%	Pass
0.115 µg/L	2	116%	Pass	72%	Pass
0.075 µg/L	1	164%	Fail	49%	Fail
0.075 µg/L	2	140%	Pass	39%	Fail

^1^ PIR—Prediction Interval of Results [21]; criteria met if 50% < PIR < 150%.

**Table 5 toxins-17-00053-t005:** Method performance of quality control samples (QCSs) in Oregon surface and finished waters during the 2022 sampling season.

QC/Water Type	Fortified Concentration	Average Recovery (%)	%RSD	Number
Low-CV	0.1 µg/L	102	12.9	51
Reagent Water	0.5 µg/L	89.0	15.8	315
QCS	0.75 µg/L	105	8.6	7
Surface Water	0.5 µg/L	99.4	19.1	74
Source Water	0.5 µg/L	93.8	17.0	238
Finished Water	0.5 µg/L	92.0	17.3	68

## Data Availability

The publicly available database containing data used to create Figure 1, Figure 2, and Table 5, as well as current data, collection times, measured concentrations, and locations, are available at the Oregon Health Authority Drinking Water Safety Online Data Portal: https://yourwater.oregon.gov/cyanocounty.php (accessed on 21 January 2025).

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
