# Peer review of "The Evaluation of a New ELISA-Based Kit for Total Microcystins as an Early Detection Tool for Microcystin Blooms in Source Waters and Its Application State-Wide to Oregon Source and Finished Drinking Waters"

_toxins, 2025, doi:10.3390/toxins17020053_

Round 1

Reviewer 1 Report

Comments and Suggestions for Authors

The subject topic of the paper is interesting. The authors did some careful investigations. But I cannot accept the paper due to the following flaws. Please find my comments: 

Any interaction study of microcystine along with various metals if the authors studied, since drinking water and other contains metal impurities.

The authors have provided the figures without showing any error bars, which are statistically not correct. They need to provide so.

Similar Microcystins Exposure Associated with drinking water was already studied. As for example: Toxins, 2023, 15(4), 293. doi: https://doi.org/10.3390/toxins15040293. Authors should provide the difference in their research while citing the important reference in their text.

The authors wrote the sentence; “ Water samples that yielded an analytical result for MCT 93 above the Minimum Reporting Level (MRL) were required to be analyzed with the more 94 specific and more sensitive liquid chromatography tandem mass spectrometer method, 95 EPA Method 544..” which need to reframe and show its potential merit in details.

The concentration of toxin level from the algal source should be mentioned. What is the purity of the commercially available ELISA kit ? What are the analytical techniques they used for the determination of the purity?

The authenticity of these data should be compared with some standard values.

Discussion on “Method Performance in Oregon Municipal and Source Waters..” should be more focussed on analysis rather than background.

Comments on the Quality of English Language

several punctuation errors and language construction should be more focused. 

Reviewer 2 Report

Comments and Suggestions for Authors

The authors evaluated the precision and accuracy of a streptavidin enhanced sensitivity (SAES) assay kit for the detection of microcystins in comparison with the standard EPA ELISA method.  The lower detection abilities of the SAES assay has the potential to alert potential contamination before the levels reach the health advisory level for small children.  This work involved two laboratories who evaluated a number of different water samples.  Based on the author’s collected data, the EPA determined the SAES assay can be used as an equivalent approved method to test for microcystins.

Although I have some background in toxins and am familiar with ELISAs and assay development, I am not an expert on EPA methods and the process of validating an equivalent (or new) assay.  For others who have limited experience with EPA methods several pieces of background information would be very helpful.

1.        There are a lot of acronyms.  In the cases where they are used only once it is probably okay to leave off the acronym.  For example, “Safe Drinking Water Act” can be spelled out without adding the acronym as it is only mentioned one time.

2.       Related, make sure to define acronyms before use

3.       Before going into the analysis, it would be helpful to orient the reader with a brief description of the methods as well as the overall structure of the study.  For example, it seems this is a competitive assay, but that could be stated explicitly.  Also, it may be useful to more explicitly state that the SAES method relies on biotinylated antibodies interacting with streptavidin-HRP versus the standard EPA method in which an anti-antibody-HRP was used to generate signal.  It would also be helpful to understand the role of the two laboratories in the evaluation of the SAES method.

4.       Unclear why Laboratory 2 did not evaluate all four spiked concentrations in Tables 2 and 3

5.       Lines 160-163 are confusing. The authors indicate that at the lowest concentration of reagent water the recovery was outside the range of 70-130%. However, from table 2 all recovery values appear to be in that range.

6.       Consider organizing as a co-mingled results and discussion as the discussion section has additional results.

Reviewer 3 Report

Comments and Suggestions for Authors

Journal: Toxins (ISSN 2072-6651)

Manuscript ID: toxins-3357538

Type: Article

Title: The Evaluation of a New ELISA-Based Kit for Total Microcystins as an Early Detection Tool for Microcystin Blooms in Source Waters and Its Application State-Wide to Oregon Source and Finished Drinking Waters

Section: Marine and Freshwater Toxins

Special Issue: Advances in Cyanotoxins: Latest Developments in Risk Assessment

This manuscript developed a new ELISA-Based Kit for Total Microcystins as an Early Detection Tool for Microcystin Blooms in Source Waters. Minimum Reporting Level (MRL) can be 0.1 µg/L, which is one third of the original MRL (0.3 µg/L) of Method 546. This is very nice.

The results are interesting. A major weakness of this manuscript is lack of details, for the concentrations, the details for filed samples. Presentation of data is not good. Also, the advantages of the SAEs assay over the EPA Method 546 are not clear.

Therefore, before this work will be recommended or will be given any possible acceptance few comments must be incorporated for improving the quality of this work as well as for further publication in this reputed journal. I have the following observations or queries and comments which may further enhance your piece of work. The authors require to modify the following points in detail.

1. Abstract

Lines 9-11

After the incident, the State of Oregon put in place drinking water rules to require routine testing of raw water, and if raw water cyanotoxin concentrations exceed trigger values, finished water testing for potentially susceptible public water systems.

Please check the grammar since “and if”.

2. Lines 17-19

“The EPA Office of Ground Water and Drinking Water (OGWDW) has concluded the SAES kit meets the requirements of Method 546. With an MRL one third of the original concentration, the new kit provides a small but critical window for early warning.”

What is your conclusion? Please delete “The EPA Office of Ground Water and Drinking Water (OGWDW) has concluded the SAES kit meets the requirements of Method 546. What is the original concentration? Is it 0.3 µg/L? If yes, please change this sentence to “With an MRL one third of the original MRL (0.3 µg/L) of Method 546, i.e. 0.1 µg/L, the new kit provides a small but critical window for early warning.”

3. Lines 20-22

“however, the new MRL has allowed Oregon to revise drinking water rules for susceptible systems to require finished water monitoring after a reduced trigger level—0.20 µg/L—is exceeded in raw water, rather than the HAL.”

What do you mean by “after a reduced trigger level—0.20 µg/L—is exceeded in raw water”. It is unclear.

4. Key Contribution

Line 26

Please insert full name of LOQ when it first occurs.

5. Section “1. Introduction”

Lines 31-36

“Since then, countless scientific investigations have yielded a much better understanding of these organisms and the toxins they produce. Today, many US states have implemented monitoring and advisory programs to detect and track harmful algal blooms (HABs), the umbrella term for all harmful blooms including algae/algal biotoxins and cyanobacteria/cyanotoxins, in recreational 35

and drinking waters.”

Please insert the references.

6. Lines 55-59

“One class of cyanotoxins, the microcystins (MCs), are hepatotoxins of varying toxicity comprising more than 200 congeners reported to date [8]. Microcystins are widespread in occurrence and have been associated with numerous global incidents [9,10]. The ability of this family of cyanotoxins to inhibit protein phosphatases gives rise to additional concerns that sublethal doses may contribute to cancer [11].”

Please insert the following paper.

Challenges of using blooms of Microcystis spp. in animal feeds: A comprehensive review of nutritional, toxicological and microbial health evaluation. https://doi.org/10.1016/j.scitotenv.2020.142319

7. Section “5. Materials and Methods”

Production of Streptavidin Enhanced Sensitivity (SAES) is not detailed. This is very important.

8. Sampling of field samples

When and where did you collect the samples? This information is useful. Readers can get concentrations for a given waterbody.

9. Section “2. Results”

How many samples did you analyze? Please present data of each sample by two bars, one for EPA Method 546, and another for the Streptavidin Enhanced Sensitivity (SAES) assay. Then readers can easily understand the data.

10. What is the maximum detection lever for the SAES kit? Please present the method for standard curves and present the figures of standard curves.

11. What is the stability or repeatability for detection of MCs for different concentrations?

12. Do you know Beacon kits for MCs and other ELISA kits? Please insert comparison with other kits, the MDL, LOD, detection range, etc.

13. Presentation of data is not good. Readers can not easily get the concentrations for the filed samples, where? Also, the advantages of the SAEs assay over the EPA Method 546.

Round 2

Reviewer 1 Report

Comments and Suggestions for Authors

The authors have revised the manuscript and this paper can be accepted.